# Intramyocardial Stem Cell Transplantation during Coronary Artery Bypass Surgery Safely Improves Cardiac Function: Meta-Analysis of 20 Randomized Clinical Trials

**DOI:** 10.3390/jcm12134430

**Published:** 2023-06-30

**Authors:** Tri Wisesa Soetisna, Ahmad Muslim Hidayat Thamrin, Diajeng Permadijana, Andi Nurul Erisya Ramadhani, Anwar Santoso, Muchtaruddin Mansyur

**Affiliations:** 1Adult Cardiac Surgery Division, Department of Thoracic and Cardiovascular Surgery, Harapan Kita National Cardiovascular Center Hospital, Jakarta 11420, Indonesia; amhtamrin@gmail.com (A.M.H.T.); diajengp99@gmail.com (D.P.); erisyanurul@gmail.com (A.N.E.R.); sugisman01@gmail.com (S.); 2Department of Thoracic and Cardiovascular Surgery, Faculty of Medicine, University of Indonesia, Jakarta 10430, Indonesia; 3Faculty of Medicine, Syarif Hidayatullah State Islamic University, Haji Hospital, Jakarta 13560, Indonesia; 4Department of Cardiology and Vascular Medicine, Harapan Kita National Cardiovascular Center Hospital, Jakarta 11420, Indonesia; awscip@gmail.com; 5Department of Cardiology and Vascular Medicine, Faculty of Medicine, University of Indonesia, Jakarta 10430, Indonesia; 6Department of Community Medicine, Faculty of Medicine, University of Indonesia, Jakarta 10430, Indonesia; muchtaruddin.mansyur@ui.ac.id

**Keywords:** stem cell, intramyocardial, coronary artery bypass, ischemic heart disease, outcome, safety

## Abstract

IMSC transplantation during CABG is considered one of the most promising methods to effectively deliver stem cells and has been widely studied in many trials. But the results of outcomes and safety of this modality still vary widely. We conducted this meta-analysis of randomized controlled trials (RCTs) to evaluate not only the outcome but also the safety of this promising method. A meta-analysis was performed according to Preferred Reporting Items for Systematic Reviews and Meta-Analysis (PRISMA) guidelines. A comprehensive literature search was undertaken using the PubMed, Scopus, and Cochrane databases. Articles were thoroughly evaluated and analyzed. Twenty publications about IMSC during CABG were included. Primary outcomes were measured using LVEF, LVESV, LVESVI, LVESD, LVEDV, LVEDVI, LVEDD, WMSI, and 6-MWT. Safety measures were depicted by total deaths, MACE, CRD, CVA, myocardial infarction, ventricular arrhythmia, and cardiac-related readmission. IMSC transplantation during CABG significantly improved LVEF (MD = 3.89%; 95% CI = 1.31% to 6.46%; *p* = 0.003) and WMSI (MD = 0.28; 95% CI = 0.01–0.56; *p* = 0.04). Most of the other outcomes showed favorable results for the IMSC group but were not statistically significant. The safety analysis also showed no significant risk difference for IMSC transplantation compared to CABG alone. IMSC during CABG can safely improve cardiac function and tend to improve cardiac volumes and dimensions. The analysis and application of influencing factors that increase patients’ responses to IMSC transplantation are important to achieve long-term improvement.

## 1. Introduction

Since its first human trial more than 20 years ago, stem cell transplantation in patients with ischemic heart disease (IHD) has been widely studied and used around the world [1,2]. Plenty of clinical trials have been conducted to evaluate the efficacy and safety of this modality [2,3,4]. This modality rose as a promising method to boost myocardial regeneration, as the myocardium is well known to have very low renewal capability, making myocardial infarction the leading worldwide cause of mortality and morbidity [3]. This disease has also trend-shifted to younger, productive-age patients lately, causing health and socio-economic problems that make prompt solutions urgently needed [5,6].

As the research on attempts to regenerate myocardial function in IHD patients develops, intramyocardial stem cell (IMSC) delivery routes have shown their superiority compared to catheter-based delivery routes or intracoronary delivery routes, especially in the most crucial variable for stem cells: retention and viability [7,8]. Intramyocardial delivery also gives an advantage in targeting viable peri-ischemic areas, as the clinician can see the target areas directly [9,10]. Studies have shown that intramyocardial delivery routes outperform other delivery routes, but they also carry bigger risks due to their invasiveness [11,12].

Because of the range of advantages of the intramyocardial delivery route compared to other routes, nowadays IMSC transplantation is the most common route to use, especially together with coronary artery bypass grafting (CABG), to reduce the risk of blind injection and gain the benefit from the bypass itself [2]. Although many studies have been conducted to evaluate the effectiveness and safety of this method, the result is still widely varied, even between big randomized controlled trials [2]. We conducted this meta-analysis of randomized controlled trials (RCTs) to evaluate not only the clinical outcome but also the safety of this method of delivery, including the data of the latest published studies to facilitate the development of further clinical usage and research into this method.

## 2. Materials and Methods

### 2.1. Eligibility Criteria

Studies used in this review were full-text randomized controlled trials on patients with ischemic heart disease comparing IMSC transplantation during CABG and CABG alone. Reviews, unpublished articles, letters to the editor, abstracts, and studies not written in English were excluded from this study. We have registered our study in PROSPERO with the registration number CRD42023401918.

### 2.2. Type of Outcome Measurement

Primary outcomes were measured using the left ventricular ejection fraction (LVEF), left ventricular end-systolic volume (LVESV), left ventricular end-systolic volume index (LVESVI), left ventricular ejection systolic diameter (LVESD), left ventricular end-diastolic volume (LVEDV), left ventricular end-diastolic volume index (LVEDVI), left ventricular ejection diastolic diameter (LVEDD), and 6 min walking test (6MWT). As secondary outcomes, we assessed the safety by assessing cardiac-related readmission, cardiac-related death, and major adverse cardiovascular events (MACE) during the follow-up period of the studies.

### 2.3. Search Methods and Identification of Studies

#### 2.3.1. Information Sources

This systematic review and meta-analysis was conducted based on the Preferred Reporting Items for Systematic Reviews and Meta-Analysis (PRISMA) guidelines [13]. The literature was acquired by searching the electronic databases PubMed, Scopus, and Embase in January–February 2022. We applied language restrictions to our search; only articles in English were selected. No publication time limitations were applied in our review.

#### 2.3.2. Search Protocol

The study question was formed using the patient/population, intervention, comparison, and outcomes (PICO) model. A predetermined PICO framework was used to select the relevant studies to be included in the review. The following keywords were used to search all trial registers and databases: (“Intramyocardial” OR “implantation” OR “application” OR “transplantation” OR “injection” OR “transepicardial” OR “transseptal” OR intramuscular) AND (“cell”) AND (“CABG” OR “Coronary artery bypass”) AND (“trial”).

### 2.4. Data Collection and Analysis

All search records were screened based on their titles and abstracts. Three authors (TWS, AMHT, DP) independently assessed the studies based on the inclusion and exclusion criteria. Those studies with irrelevant titles were excluded, followed by those with irrelevant abstracts. Non-English publications were automatically excluded. Only full-text RCT articles, fulfilling the eligibility criteria, were then assessed by all authors. The details regarding the causes of exclusion were noted and reported.

### 2.5. Data Extraction and Management

Three authors (TWS, AMHT, DP) independently extracted data from tabulations containing information on patient characteristics, treatments, research quality, and therapeutic results [14]. Details regarding the author, year of publication, study design, total patients involved, assignments of intramyocardial stem cell intervention (IMSC) and control (CABG) groups, type and dose of stem cells used, clinical outcome based on the predetermined clinical parameters (LVEF, LVESV, LVESVI, LVESD, LVEDV, LVEDVI, LVEDD, and 6MWT) and safety outcome (cardiac related readmission, cardiac-related deaths, and MACE) were summarized in a table for qualitative analysis. For measured outcomes (LVEF, LVESV, LVESVI, LVESD, LVEDV, LVEDVI, LVEDD, and 6MWT), we extracted or reanalyzed the mean difference (MD) between the experimental and control groups (with its standard deviation [SD] as reported by the study authors). Because not all studies reported mean difference between pre- and post-transplantation, we calculated the change in MD by the difference between pre- and post-transplantation MD. For the change in SD (SD_change)_, we calculated it using the following equation:$${\mathrm{SD}}_{\mathrm{change}}=\surd {{\mathrm{SD}}^{2}}_{\mathrm{baseline}}+{{\mathrm{SD}}^{2}}_{\mathrm{follow-up}}-2\;\times \;\mathrm{Corr}\;\times \;{\mathrm{SD}}_{\mathrm{baseline}}\;\times \;{\mathrm{SD}}_{\mathrm{follow-up}}$$

The SD of change in LVEF in a study by Hendrikx et al. [8] was used to calculate Corr values using the following formula:$$\mathrm{Corr}=({{\mathrm{SD}}^{2}}_{\mathrm{baseline}}+{{\mathrm{SD}}^{2}}_{\mathrm{follow-up}}-{{\mathrm{SD}}^{2}}_{\mathrm{change}})/(2\;\times \;{\mathrm{SD}}_{\mathrm{baseline}}\;\times \;{\mathrm{SD}}_{\mathrm{follow-up}})$$

The Corr value was 0.6 for both IMSC and CABG groups, and this value was used to determine the SD_change_.

For cardiac-related readmission, cardiac-related death, and MACE, we extracted the number of events in each group and analyzed the risk ratio. Four review authors (AMHT, TWS, DP, ANER) entered all data into Review Manager (RevMan) software, version 5.4 [15].

### 2.6. Risk of Bias Analysis

Each author independently assessed the risk of bias in each study included in the systematic review using the criteria outlined in the *Cochrane Handbook for Systematic Reviews of Interventions* for randomized studies, referred to as Risk of Bias 2 (RoB 2) for randomized studies [14]. The results of each interpreter’s assessment were then discussed by all of the authors. A risk-of-bias table and a summary of the bias aspects of the included studies were used to interpret the results of the systematic review in light of the overall risk-of-bias assessment [16,17].

## 3. Results

A total of 551 studies were identified and screened. Of these, 36 studies were assessed for eligibility, 20 were included in the systematic review, and all were eligible for the meta-analysis. A flowchart of the study selection processes is shown in Figure 1.

### 3.1. Study Characteristics

A total of 929 patients, 499 patients in the IMSC group, and 430 patients in the control group, were enrolled from 20 randomized clinical trial studies. Almost all studies reported predominantly male subjects in their studies (more than 70% male subjects), except the study by Komok et al. The mean age of patients was 52.1–66.4 years in the IMSC group and 55.5–66.8 years in the control group. Twelve studies used BMC/BMMC as their stem cell type for treatment, seven studies used CD133+/CD34+ cells, and three studies used hUC-MSC or ASM cells, with two studies using two types of cell in the IMSC group (hUC-MSC/BMMC and CD133+/BMMC). The mean injected stem cell dose varied from 1.42 × 10^5^ to 9.1 × 10^8^, with studies using BMC/BMMC and/or hUC-MSC/ASM seeming to have higher stem cell doses than studies using CD133+/CD34+.

Follow-up periods ranged from 4 to 60 months, with a median of 6 months. Echocardiography was used in nine studies as an outcome measurement tool, ten other studies used CMR, and one used SPECT. The summary of included studies is shown in Table 1.

### 3.2. Primary Outcome

#### 3.2.1. Left Ventricular Function, Contractility, Volume, and Dimension

A total of 787 patients from 19 studies, 433 from IMSC and 354 from the control group, underwent LVEF evaluation using follow-up time as the main indicator of left ventricle function. The heterogeneity was high (I^2^ = 82%), so a random effect model was used. Our analysis of the mean difference of LVEF change between the IMSC and control group showed a significantly higher difference in the IMSC group (MD = 3.89%; 95% CI = 1.31% to 6.46%; *p* = 0.003) (Figure 2).

Another indicator of left ventricular function and contractility, WMSI evaluation, was used in three studies including 133 patients. It also showed a significant change from baseline to the latest follow-up in the IMSC group compared to the control group (MD = 0.28; 95% CI = 0.01–0.56; p = 0.04) (Figure 3).

The analysis of left ventricular volume and dimension change showed varied results, although almost all showed a better improvement in the IMSC group. The LVESV change analysis from nine studies showed a favorable result for IMSC, but not statistically significantly (MD = −5.52 mL; 95% CI = −14.27 mL to 3.23 mL; *p* = 0.22) (Figure 4). The same result was found in the LVESVI change analysis from four studies as the favorable result for IMSC, but it was not statistically significant (MD = −1.03; 95% CI = −8.73 to 6.66; *p* = 0.79) (Figure 5). The LVESD change analysis showed a statistically significant improvement in the IMSC group compared to the control group (MD = −3.48 mm; 95% CI = −6.81 mm to −0.14 mm; *p* = 0.04) (Figure 6).

The LVEDV change analysis from 11 studies showed favorable results for IMSC, but not statistically significantly (MD = −1.43 mL; 95% CI = −9.85 mL to 6.99 mL; *p* = 0.74) (Figure 7). Otherwise, the LVEDVI analysis from four studies showed a favorable result for the control group, but not statistically significantly (MD = 1.88; 95% CI = −8.85 to 12.61; *p* = 0.73) (Figure 8). The LVEDD change analysis from three studies showed a favorable result for IMSC, but also not statistically significantly (MD = −0.92 mm; 95% CI = −5.65 mm to 3.81 mm; *p* = 0.70) (Figure 9).

#### 3.2.2. Functional Outcome

Five studies reported the results of the 6-MWT evaluation in both groups. The heterogeneity test result showed high heterogeneity (I^2^ = 91%), allowing us to use a random effect model. The mean difference analysis of 6-MWT change showed favorable results for the IMSC group, but was not statistically significant (MD = 30.07 m; 95% CI = −28.08 m to 88.22 m; *p* = 0.31) (Figure 10).

#### 3.2.3. Subgroup Analysis of LVEF

The subgroup analysis of LVEF based on the follow-up time, type of stem cells, administered stem cells dose, and baseline LVEF showed no significant difference within subgroups. The subgroup analysis of LVEF measurement methods showed a significant difference between the echo and CMR/SPECT groups (*p* = 0.04) (Table 2). The sensitivity analysis using the leave-one-out method showed that the result was not significantly affected by study exclusion.

### 3.3. Secondary Outcome (Safety Measurements)

Safety measurements of the IMSC transplantation within CABG compared to CABG alone were analyzed through various outcomes and showed no significant difference between groups. Total death events analysis from 16 studies showed a favorable result for the control group, but not statistically significantly (RR = 1.16; 95% CI = 0.60 to 2.25; *p* = 0.66) (Figure 11). MACE analysis events from 17 studies also showed a favorable result for control group, but not statistically significantly (RR = 1.19; 95% CI = 0.66 to 2.13; *p* = 0.56) (Figure 12). Myocardial infarct events analysis from 12 studies showed a favorable result for IMSC, but also not statistically significantly (RR = 0.72; 95% CI = 0.25 to 2.02; *p* = 0.53) (Figure 13). Cerebrovascular accident events analysis from 11 studies showed a favorable result for the control group, but not statistically significantly (RR = 2.12; 95% CI = 0.76 to 5.96; *p* = 0.15) (Figure 14). Cardiac-related death events analysis from 16 studies also showed a favorable result for the IMSC group, but not statistically significantly (RR = 0.94; 95% CI = 0.41 to 2.17; *p* = 0.89) (Figure 15). Cardiac-related readmission events analysis from three studies showed a favorable result for the IMSC group, but also not statistically significantly (RR = 0.78; 95% CI = 0.17 to 3.56; *p* = 0.75) (Figure 16). Ventricular arrythmia events analysis from 13 studies showed a favorable result for the control group, but not statistically significantly (RR = 1.58; 95% CI = 0.85 to 2.95; *p* = 0.15) (Figure 17).

### 3.4. Risk of Bias Analysis

A bias risk assessment of the included studies was measured using RoB2 for randomized studies, as all of the studies included were non-randomized. Overall, the risk of bias was considerably low in most of the studies. The result is shown in Figure 18.

## 4. Discussion

Twenty trials were included in our study, and no studies reported significant differences in patient characteristics between groups. Our analysis of LVEF change showed a significant LVEF improvement in the IMSC group compared to the control group, though the result was highly heterogeneous. Nevertheless, the subgroup analysis showed consistent results for most variables, indicating consistent LVEF improvement in the IMSC group. The only significant difference in the LVEF change subgroup analysis was found in the measurement method, where echocardiography evaluation showed better results than CMR/SPECT. Although significant, the result of LVEF change evaluation via echocardiography showed higher heterogeneity. This could be attributed to the echocardiography results being operator dependent, but this finding could not clinically explain the result differences. The source of heterogeneity also could not be identified appropriately, but some trials showed high LVEF deviation values, which might explain this result. A significant result of WMSI improvement in the IMSC group compared to the control in our study might confirm the result of significant LVEF improvement in the IMSC group. Significant WMSI improvement, besides being a more reliable alternative for evaluating cardiac function, also indicates the myocardial thickening of left ventricle divisions [26].

Although left ventricular function and contractility showed a significant improvement difference in IMSC compared to control, left ventricular volume and dimension evaluation showed a nonsignificant improvement difference, but most were still favorable to the IMSC group. Several studies also showed the tendency of this result [3,19,31]. This might be attributable to the time for volume and dimension improvement being more related to global reverse remodeling, which may take a longer time than the improvement of cardiac function and contractility [10,19,31]. The short follow-up time of most included studies, predominantly 6 months or less, might demonstrate the cause of the insignificant volume and dimension improvement result in the IMSC group.

The mechanism by which intramyocardial stem cell transplantation improves cardiac function outcomes in patients who underwent CABG can be seen through several different mechanisms, but each is connected to and supports other mechanisms. Like other cell-based therapies, IMSC transplantation also works primarily by the effect of secreted cytokines [3,9,10,18,19,21,25,26,27,30,31,32]. There are three main mechanisms of action of IMSC in ischemic cardiac tissue. First, by the direct effect of injected stem cells, which are multipotent stem cells that can differentiate to other cells [26,27]. The injected stem cells are thought to differentiate into cardiomyocytes and endothelial cells that can improve cardiac tissue regeneration and, later, function [3,9,21,26,27]. But this true myocardial regeneration via differentiating stem cells is still inconclusive [8,30,31]. In studies using myoblasts as stem cells, transplanted myoblasts can also directly help myocardial function due to their contractile properties [18]. Transplanted stem cells can also act as a strengthening scaffold for the myocardial tissue [18]. Second, the paracrine effect of cytokines acts on resident stem cells, hibernating myocardial tissue, and the extracellular matrix in ischemic and peri-ischemic myocardial tissue that stimulates repair, suppressing the process of myocardial fibrosis and hypertrophy, then attenuating myocardial remodeling [3,25,26,27,30,31,32]. This leads to left ventricular (LV) reverse remodeling, then leads to a global improvement in LV function [26,30]. Third, paracrine effects on ischemic and peri-ischemic areas also affect myocardial vessels by improving coronary microvascular dysfunction, amplifying the angiogenic response (angiogenesis), including collateral vessel formation [3,9,18,21,24,25,27,30,31]. This ensures long-term perfusion control and stability on that ischemic and peri-ischemic area [3,27,31]. In the end, this leads to a long-term and broader LV reverse remodeling [3,22,27]. These three mechanisms together result in the improvement of cardiac function. A summary of the stem cell mechanisms in improving outcomes is illustrated in Figure 19, below.

Although promising, the results of the studies included were still inconsistent. Fortunately, these studies elaborate on factors that may contribute to a patient’s response to IMSC. First, patients’ responses to IMSC differ widely, and the responders have shown an improvement even in the relatively short follow-up period [8,32]. This improvement in the responders will continue over time in a long-term period, while not for non-responders [33]. Low LV volume and diameter levels contributed to the responder group, as these levels are believed to be determined by the extent of viable myocardial cells [22,32,33]. A previous study showed that LVESVI 70 mL/m^2^ was the cut-off point for determining whether a patient would respond to stem cell-based treatment or not [33]. Lower body weight, lower circulating angiogenic factors, and higher bone marrow stem cell levels are also associated with the responder group [8,32]. The proposed potential mechanism of this non-responder group was impaired angiogenesis caused by a dysfunctional bone marrow response [8,32].

Stem cell type is also postulated to affect patients’ responses to IMSC. Homogenous differentiated cell populations such as CD34+ and CD 133+ are more attributable to angiogenesis, thus improving cardiac function [20,25]. A study by Hendrikx et al. showed that CD34+ levels were significantly higher in the responders [8]. MSC may have a longer paracrine effect than BMMC [19]. Transplantation timing may also significantly increase patients’ responses, where the subacute phase is considered the best [23]. In the subacute phase, the cytokine level was high enough to induce tissue regeneration but not as high as in the acute phase, which could affect stem cell engraftment and survival. Also in this phase, no significant fibrosis like that was found in the chronic phase that could interfere with the myocardial tissue reverse remodeling [23,28].

Several intraoperative techniques are also considered to increase patients’ responses to IMSC. Standard myocardial injection to the contractile area may cause stem cell loss and can be encountered by the fibrin glue sealing [25]. The use of collagen hydrogel as an injection vehicle is also considered to increase cell retention [4]. The selection of the injection area is also important, where areas of peri-infarction with a range of viable hibernating myocardium are considered to be best [9,10,31]. The ventricular septum plays an important role in ventricular contractility, and thus the ischemia in this area will affect ventricular function [3]. A study by Nasseri et al. showed that insignificant LV function improvement may result from the inaccessibility of the septum for stem cell injection [28]. But our previous study showed that transseptal injection was still applicable by injecting stem cells through radiologically measured points along the left anterior descending artery, which resulted in significant LVEF and WMSI improvement after 6 months [3].

Safety issues remain a major consideration of IMSC. Our analysis showed no significantly increased risk of death or other adverse events related to IMSC. Arrhythmia is considered one of the biggest risks of IMSC because of its invasive nature to the myocardial tissue [8,27,30]. Although the results of our analysis showed that the risk of arrhythmia was higher in the IMSC group, the risk of cardiac-related death was lower, showing that the arrhythmias were non-lethal. In addition, the use of purified homogenous stem cells can lower the risk of arrhythmia [31]. This gives us a view that IMSC transplantation during CABG is considered safe.

### Limitations

Several limitations apply to our study. First, there was high heterogeneity in the trial results because of many factors, including a low number of patients and a high number of drop out in some trials. Second, the assessed parameter remained inconsistent in some studies, so some measured outcomes could only be obtained from a small number of trials. The follow-up period was also inconsistent, and most of the trials only reported a short-term follow-up, which reduced the value of outcome improvement that could be assessed as we have seen that the outcome improvement of successful transplantation increased over time. This also can affect the interpretation of the results, as the reported outcomes were heterogeneous and could not be evaluated with more precise results.

## 5. Conclusions

IMSC during CABG can safely improve cardiac function. It also shows a tendency to improve cardiac volumes and dimensions over a long-term period. The analysis and application of influencing factors that increase patients’ responses to IMSC transplantation are important to achieve long-term continuous benefits from stem cells, and these factors have to be studied in more depth. More well-designed clinical trials with long-term outcome assessments are required to confirm these results.

## Figures and Tables

**Figure 1 jcm-12-04430-f001:**
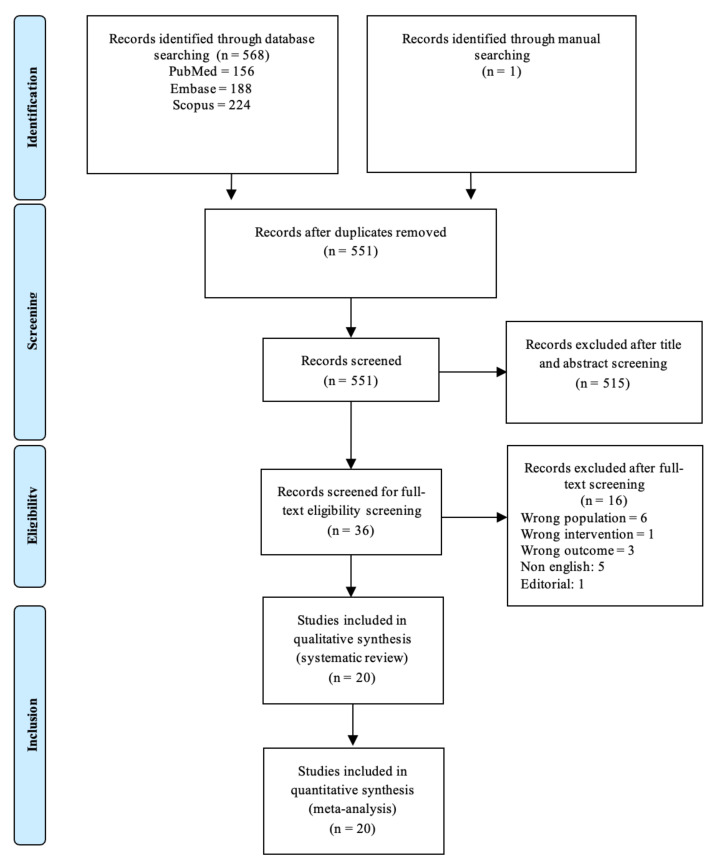
Preferred Reporting Items for Systematic Reviews and Meta-Analysis (PRISMA) guidelines flowchart.

**Figure 2 jcm-12-04430-f002:**
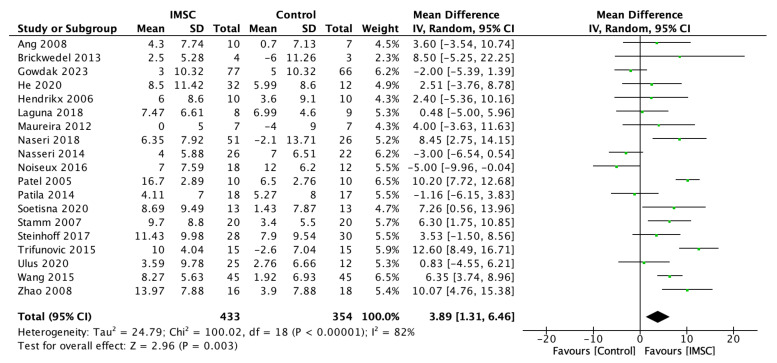
Forest plot of LVEF change (%) difference between ISMC and control group [3,4,8,9,10,18,19,20,21,23,24,25,26,27,28,29,30,31,32].

**Figure 3 jcm-12-04430-f003:**
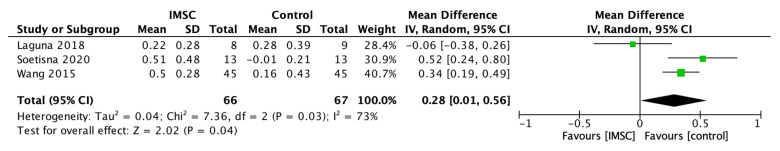
Forest plot of WMSI change difference between ISMC and control group [3,23,26].

**Figure 4 jcm-12-04430-f004:**
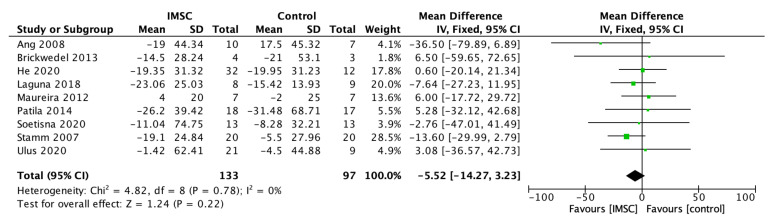
Forest plot of LVESV (mL) change difference between ISMC and control group [3,4,18,19,20,23,24,25,31].

**Figure 5 jcm-12-04430-f005:**
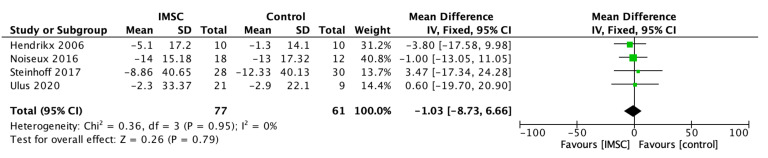
Forest plot of LVESVI change difference between ISMC and control group [8,19,29,32].

**Figure 6 jcm-12-04430-f006:**
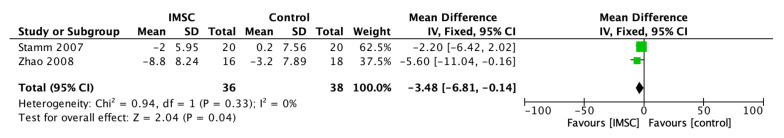
Forest plot of LVESD (mm) change difference between ISMC and control group [27,31].

**Figure 7 jcm-12-04430-f007:**
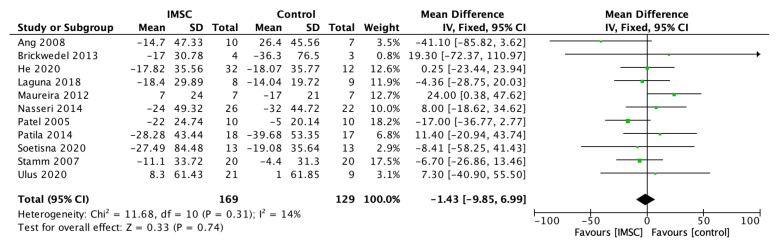
Forest plot of LVEDV (mL) change difference between ISMC and control group [3,4,18,19,20,23,24,25,28,30,31].

**Figure 8 jcm-12-04430-f008:**
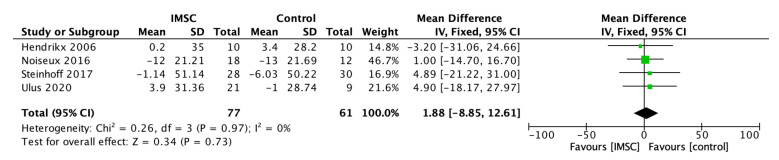
Forest plot of LVEDVI change difference between ISMC and control group [8,19,29,32].

**Figure 9 jcm-12-04430-f009:**
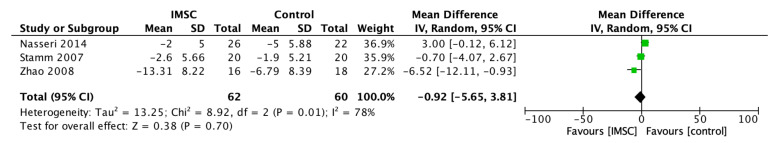
Forest plot of LVEDD (mm) change difference between ISMC and control group [27,28,31].

**Figure 10 jcm-12-04430-f010:**
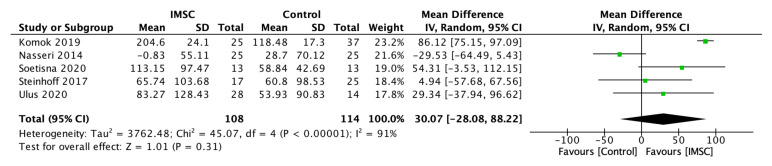
Forest plot of 6-MWT change difference between ISMC and control group [3,19,22,28,32].

**Figure 11 jcm-12-04430-f011:**
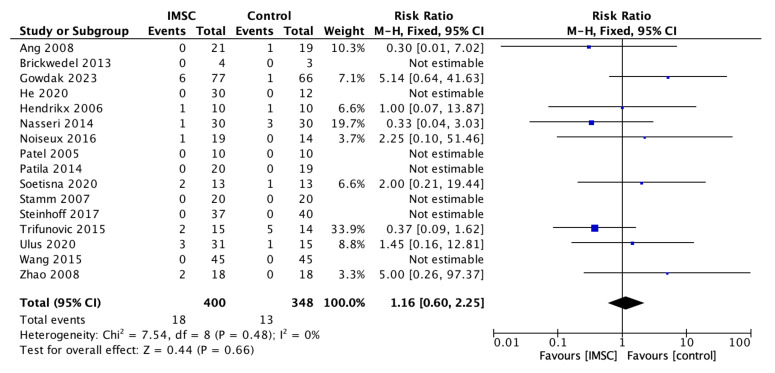
Forest plot of total death incidence difference between ISMC and control group [3,4,10,11,18,19,20,21,25,26,27,28,29,30,31,32].

**Figure 12 jcm-12-04430-f012:**
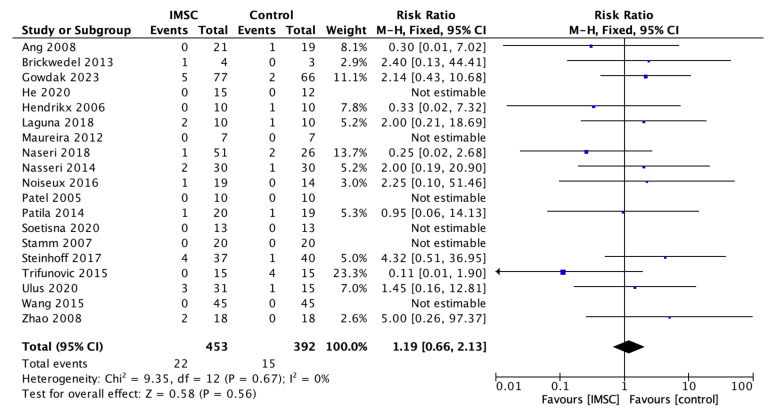
Forest plot of MACE incidence difference between ISMC and control group [3,4,8,9,10,18,19,20,21,23,24,25,26,27,28,29,30,31,32].

**Figure 13 jcm-12-04430-f013:**
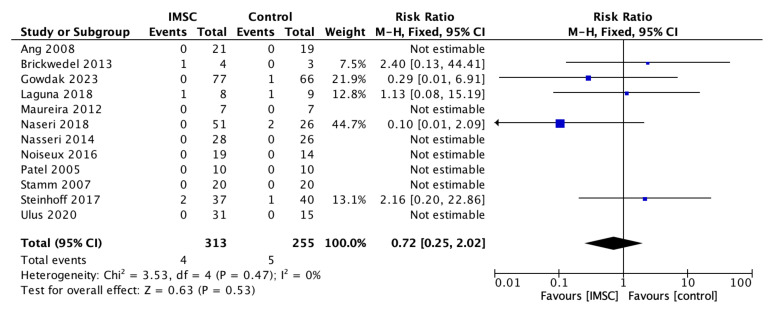
Forest plot of myocardial infarct incidence difference between ISMC and control group [3,4,10,11,18,19,20,21,23,25,26,27,28,29,30,31,32].

**Figure 14 jcm-12-04430-f014:**
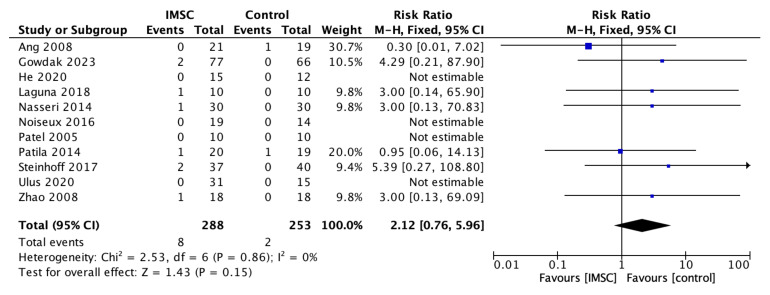
Forest plot of cerebrovascular incidence difference between ISMC and control group [4,19,20,21,23,25,27,28,29,30,32].

**Figure 15 jcm-12-04430-f015:**
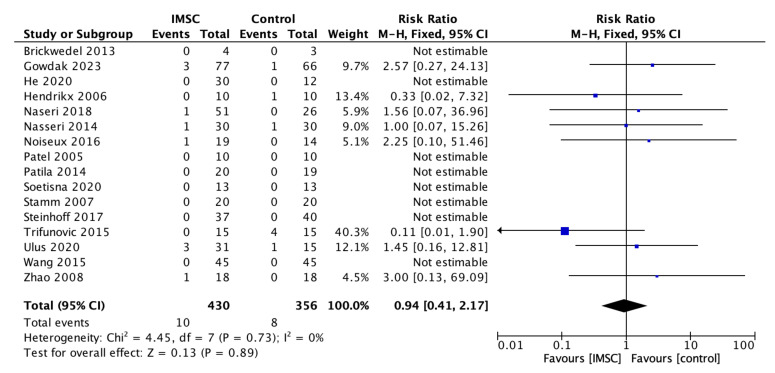
Forest plot of cardiac-related death incidence difference between ISMC and control group [3,4,8,9,10,18,19,21,25,26,27,28,29,30,31,32].

**Figure 16 jcm-12-04430-f016:**
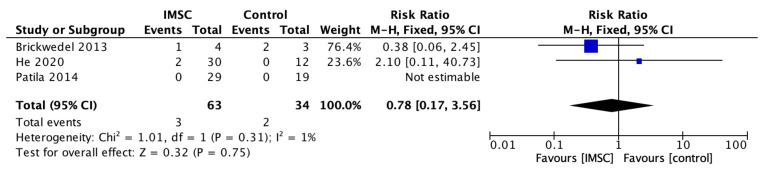
Forest plot of cardiac-related readmission incidence difference between ISMC and control group [4,18,25].

**Figure 17 jcm-12-04430-f017:**
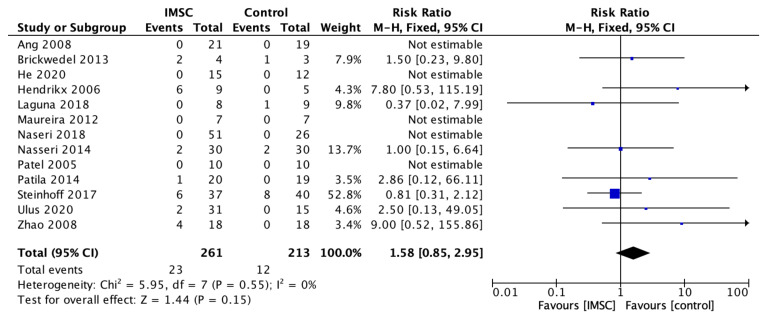
Forest plot of ventricular arrhythmia incidence difference between ISMC and control group [3,4,8,18,19,20,23,24,25,27,28,30,31].

**Figure 18 jcm-12-04430-f018:**
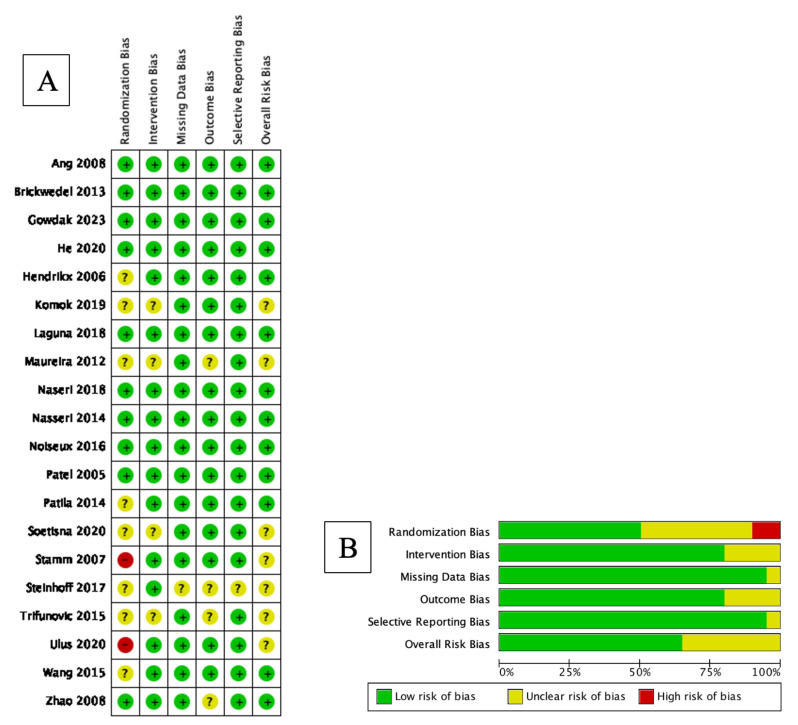
Results of RoB2 assessment. (**A**) Risk of bias summary: review authors’ judgments about each risk of bias item for each included study using RoB2 for randomized studies. (**B**) Risk of bias graph: review authors’ judgments about each risk of bias item presented as percentages across all included studies using RoB2 for randomized studies [3,4,8,9,10,18,19,20,21,22,23,24,25,26,27,28,29,30,31,32].

**Figure 19 jcm-12-04430-f019:**
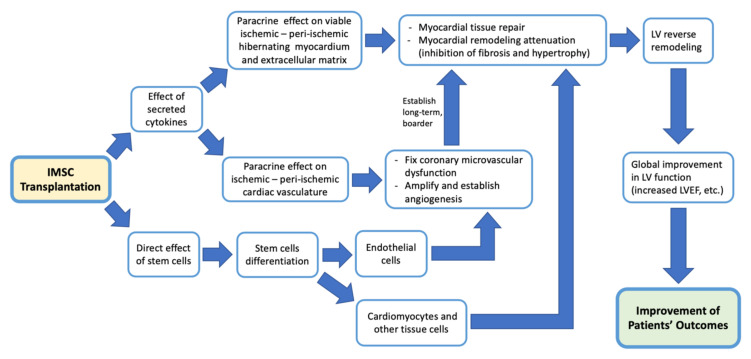
How IMSC transplantation improves patient’s outcomes. Transplanted stem cells can directly differentiate into many kinds of tissues, including cardiomyocytes and endothelial cells [3,9,21,26,27]. This in turn supports myocardial tissue and vasculature repair [3,9,21,26,27]. The transplanted stem cells also release cytokines that exert paracrine effects [3,25,26,27,30,31,32]. Paracrine effects on damaged but viable and hibernating myocardial tissue lead to tissue repair and remodeling attenuation, resulting in LV reverse remodeling [26,30]. This is supported by the paracrine effect on cardiac vessels, which stimulates vasculature repair and angiogenesis, increasing myocardial tissue perfusion and ensuring continuous LV reverse remodeling [3,27,31]. All of these mechanisms together will lead to global LV function improvement, which will result in an overall improvement of patients’ outcomes.

**Table 1 jcm-12-04430-t001:** Summary of the studies included in the review.

Author, Year	Country	Sample		Age (Years, Mean +/− SD)		SC Type	SC Dose (Mean +/− SD)	Control Group Treatment	Follow Up Period (Months)	Outcome Measurement Method
		IMSC (Male, *n*)	Cont. (Male, *n*)	IMSC	Cont.					
ASM/hUC-MSC										
Brickwedel et al., 2013 [18]	Germany	4	3	56.5 ± 5.1	62 ± 6.6	ASM	High dose: 800 × 10^6^ Low dose: 400 × 10^6^	CABG + Placebo	6	Echo
He et al., 2020 [4]	China	32; 16/16 (25)	12 (7)	61.6 8.4	65.2 7.9	hUC-MSC (±collagen hydrogel)	10^8^	CABG only	12	CMR
Ulus et al., 2020 [19]	Turkey	37; 25/12 (37)	16 (16)	60.2 3	65.3 1.7	hUC-MSC/BMMC	21–26 × 10^6^/70 × 10^7^	CABG only	12	Echo/CMR
BMC/BMMC										
Ang et al., 2008 [20]	United Kingdom	21 (15)	20 (18)	64.7 ± 8.7	61.3 ± 8.3	BMC	BMC 84 ± 56 × 10^6^ CD34+/CD117+ cells 142 ± 166 × 10^3^	CABG only	6	CMR
Gowdak et al., 2023 [21]	Brazil	77 (67)	66 (51)	59 11	57 11	BMC	Min 10^8^	CABG + Placebo	12	Echo
Hendrikx et al., 2006 [8]	Belgium	10 (10)	10 (7)	63.2 8.5	66.8 9.2	BMC	60.25 × 10^6^ ± 31.35 × 10^6^	CABG only	4	CMR
Komok et al., 2019 [22]	Russia	25 (6)	37 (4)	61 8	61.7 6.8	BMMC	4.06 x 10^6^ ± 1.78 x 10^6^	CABG + Placebo	12	Echo
Laguna et al., 2018 [23]	Spain	8 (7)	9 (8)	62.6 8.4	64.8 11.5	BMMC	10^7^	CABG + Placebo	9	CMR
Maureira et al., 2012 [24]	France	7 (7)	7 (6)	58 10	57 10	BMMC	30–40 × 10^7^	CABG only	6	CMR
Naseri et al., 2018 [9]	Iran	51; 30/21 (46)	26 (23)	52.1 7.9	55.5 8.5	BMMC/CD133+	BMMC 564.63 ± 69.35 × 10^6^ CD133 + 8.19 ± 4.26 × 10^6^	CABG + Placebo	18	SPECT
Patila et al., 2014 [25]	Finland	20 (19)	19 (18)	65 12.8	64 9.6	BMMC	9.1 ± 6.6 × 10^8^	CABG + Placebo	12	CMR
Trifunovic et al., 2015 [10]	Serbia	15 (14)	15 (14)	53.8 10.1	60 6.8	BMMC	70.7 ± 32.4 × 10^6^	CABG only	60	Echo
Wang et al., 2015 [26]	China	45 (37)	45 (35)	61.4 7.4	62.9 6.9	BMC	5.21 ± 0.44 × 10^8^	CABG + Placebo	6	Echo
Zhao et al., 2008 [27]	China	18 (15)	18 (15)	60.3 10.4	59.1 15.7	BMMC	6.6 ± 5.12 × 10^8^	CABG + Placebo	6	Echo
CD34+/CD133+										
Nassseri et al., 2014 [28]	Germany	30 (28)	30 (29)	61.9 7.3	62.7 10.6	CD133+	5.8 ± 4.7 × 10^6^	CABG + Placebo	6	CMR
Noiseux et al., 2016 [29]	Canada	19 (17)	14 (13)	66.4 6.5	63.1 7.2	CD133+	6.5 ± 3.1 × 10^6^	CABG + Placebo	6	CMR
Patel et al., 2005 [30]	Argentina	10 (8)	10 (8)	64.8 3.9	63.6 4.9	CD34+	22 × 10^6^ (median)	CABG only	6	Echo
Soetisna et al., 2020 [3]	Indonesia	13 (12)	13 (12)	54.6 8.1	57.5 8.3	CD133+	5–10 × 10^6^	CABG only	6	CMR
Stamm et al., 2007 [31]	Germany	20 (15)	20 (16)	62 10.2	63.5 8.4	CD133+	5.5 ± 1.9 × 10^6^	CABG only	6	Echo
Steinhoff et al., 2017 [32]	Germany	37 (33)	40 (34)	63.5 8.3	62.9 8.5	CD133+	2.3 ± 1.4 × 10^6^	CABG + Placebo	6	CMR

ASM: autologous skeletal myoblast; BMC: bone marrow cells; BMMC: bone marrow mononuclear cells; CABG: coronary artery bypass graft; CMR: cardiac magnetic resonance; Cont.: control; hUC-MSC: human umbilical cord—mesenchymal stem cell; IMSC: intramyocardial stem cell; SC: stem cell; SPECT: single-photon emission computerized tomography.

**Table 2 jcm-12-04430-t002:** Subgroup analysis of LVEF change.

Subgroup Variable	No. of Studies	MD (95% CI)	Heterogeneity (I^2^) (%)	*p*-Value
LVEF Follow-Up Time				
≤6 months	12	4.38 (1.18–7.58)	81	0.65
>6 months	7	3.10 (−1.47–7.67)	84	
LVEF Measurement Method				
Echo	8	6.47 (2.76–10.19)	85	0.04
CMR/SPECT	11	1.67 (−0.96–4.30)	58	
Type of Stem Cells				
BMC/BMMC	11	4.40 (1.19–7.62)	77	0.50
CD133+/CD34+	7	4.13 (−0.92–9.19)	89	
ASM/hUC-MSC	3	1.45 (−2.54–5.44)	0	
Mean Administered Stem Cell Dose				
<10^8^	12	4.75 (1.48–8.02)	79	0.49
≥10^8^	9	3.05 (−0.51–6.61)	79	
Mean Baseline LVEF				
<35%	7	4.37 (−0.52–9.25)	84	0.81
≥35%	12	3.65 (0.46–6.84)	82	

## Data Availability

No new data were created or analyzed in this study. Data sharing is not applicable to this article.

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
