# Peer review of "Intramyocardial Stem Cell Transplantation during Coronary Artery Bypass Surgery Safely Improves Cardiac Function: Meta-Analysis of 20 Randomized Clinical Trials"

_jcm, 2023, doi:10.3390/jcm12134430_

Round 1

Reviewer 1 Report

The manuscript titled "Intramyocardial Stem Cell Transplantation during Coronary 2 Artery Bypass Surgery Safely Improves Cardiac Function: 3 Meta-analysis of 20 Randomized Clinical Trials" reviewed articles which perforemd ste cell therapy in patients undergoing CABG and and put together the efficacy of stem cell therapy for patients undergoing CABG. 

The article is well written and is very relevant to the clinical research. The studies on using stem cells and their efficacy is scattered and studies like this looking at the data from several studies and making a conclusion is highly commentable. 

Author Response

Response to Reviewer 1 Comments

Point 1: 

The manuscript titled "Intramyocardial Stem Cell Transplantation during Coronary 2 Artery Bypass Surgery Safely Improves Cardiac Function: 3 Meta-analysis of 20 Randomized Clinical Trials" reviewed articles which performed stem cell therapy in patients undergoing CABG and put together the efficacy of stem cell therapy for patients undergoing CABG. 

The article is well-written and is very relevant to clinical research. The studies on using stem cells and their efficacy are scattered and studies like this looking at the data from several studies and making a conclusion is highly commentable.

Response 1:

Thank you for the review of our manuscript. We greatly value the time and effort you put into reviewing the manuscript. We hope our study gives novelty to stem cell studies, especially for stem cell implantation during surgical revascularization (CABG). 

Reviewer 2 Report

1. What is the novelty of the current review as compared to others previously published in the literature?

2. The N is somewhat low for anything but large effects to be observed.  Small but biological effects may go undetected.

3. Please discuss the long-term effects of the therapy for patients (longer than 6 months).  Possibly its too soon for the review without longer timepoints.

4. Introduction section is too short for a review.

5. Please list the titles and ID# for the 20 studies chosen for analysis.

6. In Table 1, please group the studies by stem cell source in order to facilitate comparisons.  Also, if possible, please analyze the effect of stem cell dose and type.

7. In the analyses many of the outcomes were favorable but not statistically significant.  Was this possibly due to low cell doses used?

8. Were there any functional biological effects of the therapy such as increased walking distance, etc.?

Author Response

Point 1: What is the novelty of the current review as compared to others previously published in the literature?

 Response 1: Our study is the first study to examine the intramyocardial stem cell delivery method exclusively for CABG patients. Several studies and reviews before assessed the efficacy of stem cell therapy during CABG, but not exclusively for intramyocardial delivery. Also, to the best of our knowledge, none of them focus on the safety of stem cell therapy as we focused on this study.

We also elaborate and conclude the proposed mechanism for how intramyocardial stem cell therapy can improve patient outcomes (Figure 19), together with the factors that may contribute to a patient’s response to intramyocardial stem cell therapy

Point 2: The N is somewhat low for anything but large effects to be observed. Small but biological effects may go undetected. 

Response 2: The number of studies for the several variables assessed is indeed not too large, but the number of patients for each variable is quite high (an average of more than 100 samples per group), so in our opinion, it is quite good. Nevertheless, one of the limitations that we have included and are aware of from this study is that the assessed parameters remain inconsistent in some studies so some measured outcomes can only be obtained from a small number of trials.

Point 3: Please discuss the long-term effects of the therapy for patients (longer than 6 months). Possibly it's too soon for the review without longer time points.

 Response 3: We have included the discussion for the long-term effects of the therapy in line number 318 – 326. Our analysis shows that the time for cardiac volume and dimension improvement is more related to global reverse remodeling which may take longer time than the improvement of cardiac function and contractility. The short follow-up time of most included studies, dominantly 6 months or less, might demonstrate the cause of the insignificant volume and dimension improvement result in the IMSC group. But the 6 months' time is enough to improve cardiac function, depicted by LVEF and WMSI.

Point 4: Introduction section is too short for a review.

Response 4: We have added some in the introduction section.

Point 5: Please list the titles and ID# for the 20 studies chosen for analysis.

 Response 5: Here are the titles and ID# for the 20 studies chosen for analysis.

  1. Ang, K.L.; Chin, D.; Leyva, F.; Foley, P.; Kubal, C.; Chalil, S.; Srinivasan, L.; Bernhardt, L.; Stevens, S.; Shenje, L.T.; et al. Randomized, Controlled Trial of Intramuscular or Intracoronary Injection of Autologous Bone Marrow Cells into Scarred Myocardium during CABG versus CABG Alone. Clin. Pract. Cardiovasc. Med. 2008, 5, 663–670, doi:10.1038/ncpcardio1321
  2. Brickwedel, J.; Gulbins, H.; Reichenspurner, H. Long-Term Follow-up after Autologous Skeletal Myoblast Transplantation in Ischaemic Heart Disease. Cardiovasc. Thorac. Surg. 2014, 18, 61–66, doi:10.1093/icvts/ivt434.
  3. Gowdak, L.H.W.; Schettert, I.T.; Rochitte, C.E.; de Carvalho, L.P.; Vieira, M.L.C.; Dallan, L.A.O.; de Oliveira, S.A.; César, L.A.M.; Brito, J.O.R.; Guarita-Souza, L.C.; et al. Additional Improvement in Regional Myocardial Ischemia after Intracardiac Injection of Bone Marrow Cells during CABG Surgery. Cardiovasc. Med. 2023, 10, 1–8, doi:10.3389/fcvm.2023.1040188
  4. He, X.; Wang, Q.; Zhao, Y.; Zhang, H.; Wang, B.; Pan, J.; Li, J.; Yu, H.; Wang, L.; Dai, J.; et al. Effect of Intramyocardial Grafting Collagen Scaffold With Mesenchymal Stromal Cells in Patients With Chronic Ischemic Heart Disease. JAMA Netw. Open 2020, 3, e2016236, doi:10.1001/jamanetworkopen.2020.16236
  5. Hendrikx, M.; Hensen, K.; Clijsters, C.; Jongen, H.; Koninckx, R.; Bijnens, E.; Ingels, M.; Jacobs, A.; Geukens, R.; Dendale, P.; et al. Recovery of Regional but Not Global Contractile Function by the Direct Intramyocardial Autologous Bone Marrow Transplantation. Circulation 2006, 114, doi:10.1161/CIRCULATIONAHA.105.000505
  6. Komok, V. V.; Bunenkov, N.S.; Beliy, S.A.; Pizin, V.M.; Kondratev, V.M.; Dulaev, A. V.; Kobak, A.E.; Maksimova, T.S.; Sergienko, I.P.; Parusova, E. V.; et al. Evaluation of Effectiveness of Combined Treatment of Coronary Heart Disease - Coronary Artery Bypass Grafting, Transplantation of Autologous Bone Marrow Mononuclear Cells: A Randomized, Blind Placebo-Controlled Study. Transplantologii i Iskusstv. Organov 2019, 21, 54–66, doi:10.15825/1995-1191-2019-4-54-66
  7. Laguna, G.; Di Stefa No, S.; Maroto, L.; Fulquet, E.; Echevarría, J.R.; Revilla, A.; Urueña, N.; Sevilla, T.; Arnold, R.; Ramos, B.; et al. Effect of Direct Intramyocardial Autologous Stem Cell Grafting in the Sub-Acute Phase after Myocardial Infarction. Cardiovasc. Surg. (Torino).2018, 59, 259–267, doi:10.23736/S0021-9509.17.10126-6
  8. Maureira, P.; Tran, N.; Djaballah, W.; Angioï, M.; Bensoussan, D.; Didot, N.; Fay, R.; Sadoul, N.; Villemot, J.P.; Marie, P.Y. Residual Viability Is a Predictor of the Perfusion Enhancement Obtained with the Cell Therapy of Chronic Myocardial Infarction: A Pilot Multimodal Imaging Study. Nucl. Med. 2012, 37, 738–742, doi:10.1097/RLU.0b013e318251e38a
  9. Naseri, M.H.; Madani, H.; Tafti, S.H.A.; Farahani, M.M.; Saleh, D.K.; Hosseinnejad, H.; Hosseini, S.; Hekmat, S.; Ahmadi, Z.H.; Dehghani, M.; et al. Erratum: COMPARE CPM-RMI Trial: Intramyocardial Transplantation of Autologous Bone Marrow-Derived CD133+ Cells and MNCs during CABG in Patients with Recent MI: A Phase II/III, Multicenter, Placebo-Controlled, Randomized, Double-Blind Clinical Trial. (Cell. Cell J. 2018, 20, 449, doi:10.22074/cellj.2018.6018
  10. Nasseri, B.A.; Ebell, W.; Dandel, M.; Kukucka, M.; Gebker, R.; Doltra, A.; Knosalla, C.; Choi, Y.H.; Hetzer, R.; Stamm, C. Autologous CD133+ Bone Marrow Cells and Bypass Grafting for Regeneration of Ischaemic Myocardium: The Cardio133 Trial. Heart J. 2014, 35, 1263–1274, doi:10.1093/eurheartj/ehu007
  11. Noiseux, N.; Mansour, S.; Weisel, R.; Stevens, L.M.; Der Sarkissian, S.; Tsang, K.; Crean, A.M.; Larose, E.; Li, S.H.; Wintersperger, B.; et al. The IMPACT-CABG Trial: A Multicenter, Randomized Clinical Trial of CD133+ Stem Cell Therapy during Coronary Artery Bypass Grafting for Ischemic Cardiomyopathy. Thorac. Cardiovasc. Surg. 2016, 152, 1582-1588.e2, doi:10.1016/j.jtcvs.2016.07.067
  12. Patel, A.N.; Geffner, L.; Vina, R.F.; Saslavsky, J.; Urschel, H.C.; Kormos, R.; Benetti, F. Surgical Treatment for Congestive Heart Failure with Autologous Adult Stem Cell Transplantation: A Prospective Randomized Study. Thorac. Cardiovasc. Surg. 2005, 130, 1631-1638.e2, doi:10.1016/j.jtcvs.2005.07.056
  13. Pätilä, T.; Lehtinen, M.; Vento, A.; Schildt, J.; Sinisalo, J.; Laine, M.; Hämmäinen, P.; Nihtinen, A.; Alitalo, R.; Nikkinen, P.; et al. Autologous Bone Marrow Mononuclear Cell Transplantation in Ischemic Heart Failure: A Prospective, Controlled, Randomized, Double-Blind Study of Cell Transplantation Combined with Coronary Bypass. Hear. Lung Transplant. 2014, 33, 567–574, doi:10.1016/j.healun.2014.02.009
  14. Soetisna, T.W.; Sukmawan, R.; Setianto, B.; Mansyur, M.; Murni, T.W.; Listiyaningsih, E.; Santoso, A. Combined Transepicardial and Transseptal Implantation of Autologous CD 133+ Bone Marrow Cells during Bypass Grafting Improves Cardiac Function in Patients with Low Ejection Fraction. Card. Surg. 2020, 35, 740–746, doi:10.1111/jocs.14454
  15. Stamm, C.; Kleine, H.D.; Choi, Y.H.; Dunkelmann, S.; Lauffs, J.A.; Lorenzen, B.; David, A.; Liebold, A.; Nienaber, C.; Zurakowski, D.; et al. Intramyocardial Delivery of CD133+ Bone Marrow Cells and Coronary Artery Bypass Grafting for Chronic Ischemic Heart Disease: Safety and Efficacy Studies. Thorac. Cardiovasc. Surg. 2007, 133, doi:10.1016/j.jtcvs.2006.08.077
  16. Steinhoff, G.; Nesteruk, J.; Wolfien, M.; Kundt, G.; Börgermann, J.; David, R.; Garbade, J.; Große, J.; Haverich, A.; Hennig, H.; et al. Cardiac Function Improvement and Bone Marrow Response –: Outcome Analysis of the Randomized PERFECT Phase III Clinical Trial of Intramyocardial CD133+ Application After Myocardial Infarction. EBioMedicine 2017, 22, 208–224, doi:10.1016/j.ebiom.2017.07.022
  17. Trifunović, Z.; Obradović, S.; Balint, B.; Ilić, R.; Vukić, Z.; Šišić, M.; Kostić, J.; Rusović, S.; Dobrić, M.; Ostojić, G. Funkcionalni Oporavak Bolesnika Sa Ishemijskom Kardiomiopatijom Lečenih Implantacijom Mononuklearnih Ćelija Koštane Srži Tokom Aortokoronarne Bajpas Hirurgije. Pregl. 2015, 72, 225–232, doi:10.2298/VSP140109071T
  18. Ulus, A.T.; Mungan, C.; Kurtoglu, M.; Celikkan, F.T.; Akyol, M.; Sucu, M.; Toru, M.; Gul, S.S.; Cinar, O.; Can, A. Intramyocardial Transplantation of Umbilical Cord Mesenchymal Stromal Cells in Chronic Ischemic Cardiomyopathy: A Controlled, Randomized Clinical Trial (HUC-HEART Trial). J. Stem Cells 2020, 13, 364–376, doi:10.15283/IJSC20075
  19. Wang, H.; Wang, Z.; Jiang, H.; Ma, D.; Zhou, W.; Zhang, G.; Chen, W.; Huang, J.; Liu, Y. Effect of Autologous Bone Marrow Cell Transplantation Combined with Off-Pump Coronary Artery Bypass Grafting on Cardiac Function in Patients with Chronic Myocardial Infarction. 2015, 130, 27–33, doi:10.1159/000369381
  20. Zhao, Q.; Sun, Y.; Xia, L.; Chen, A.; Wang, Z. Randomized Study of Mononuclear Bone Marrow Cell Transplantation in Patients With Coronary Surgery. Thorac. Surg. 2008, 86, 1833–1840, doi:10.1016/j.athoracsur.2008.08.068

Point 6: In Table 1, please group the studies by stem cell source in order to facilitate comparisons.  Also, if possible, please analyze the effect of stem cell dose and type

 Response 6: We have grouped the studies by stem cell source in Table 1. We have analyzed the effect of stem cell dose and type and also other factors in a subgroup analysis (result section 3.2.3, line number 235 - 243; and discussion section, line number 305-317 and 377 - 381)

Point 7: In the analyses, many of the outcomes were favorable but not statistically significant. Was this possibly due to the low cell doses used?

 Response 7: The non-statistically significant outcomes were outcomes that describe cardiac volumes and dimensions. We have analyzed and discussed this result in line number 318-326. It may result from the fact that the time for cardiac volume and dimension improvement is more related to global reverse remodeling which may take longer time than the improvement of cardiac function and contractility. The short follow-up time of most included studies, dominantly 6 months or less, might demonstrate the cause of the insignificant volume and dimension improvement result in the IMSC group. The result from the studies included show that stem cell doses didn’t significantly affect this result, as the cell doses in included studies were already measured before trial and the doses were more likely related to stem cell type (homogenous differentiated stem cells have lower doses than heterogeneous stem cells, etc)

Point 8: Were there any functional biological effects of the therapy such as increased walking distance, etc.?

Response 8: Yes, the stem cells therapy will have functional biological effects on the patients. We discussed this in line 327 – 363, including figure 19. Stem cells therapy will lead to global LV function improvement which will result in overall improvement of patients’ outcomes, including functional effects.